# A Machine Learning Approach to Finding the Fastest Race Course for Professional Athletes Competing in Ironman^®^ 70.3 Races between 2004 and 2020

**DOI:** 10.3390/ijerph20043619

**Published:** 2023-02-17

**Authors:** Mabliny Thuany, David Valero, Elias Villiger, Pedro Forte, Katja Weiss, Pantelis T. Nikolaidis, Marília Santos Andrade, Ivan Cuk, Caio Victor Sousa, Beat Knechtle

**Affiliations:** 1Faculty of Sports, University of Porto, 4200-450 Porto, Portugal; 2Ultra Sports Science Foundation, 69310 Pierre-Benite, France; 3Klinik für Allgemeine Innere Medizin, Kantonsspital St. Gallen, 9000 St. Gallen, Switzerland; 4CI-ISCE, Higher Institute of Educational Sciences of the Douro, 4560-708 Penafiel, Portugal; 5Research Center in Sports, Health and Human Development, 6201-001 Covilhã, Portugal; 6Department of Sport Sciences, Instituto Politécnico de Bragança, 5300-253 Bragança, Portugal; 7Institute of Primary Care, University Hospital Zurich, 8091 Zurich, Switzerland; 8School of Health and Caring Sciences, University of West Attica, 12243 Athens, Greece; 9Department of Physiology, Federal University of São Paulo, São Paulo 04021-001, Brazil; 10Faculty of Sport and Physical Education, University of Belgrade, 11000 Belgrade, Serbia; 11Health and Human Sciences, Loyola Marymount University, Los Angeles, CA 90045, USA; 12Medbase St. Gallen Am Vadianplatz, 9001 St. Gallen, Switzerland

**Keywords:** endurance, cycling, half-distance Ironman, swimming, triathlon, running

## Abstract

Our purpose was to find the fastest race courses for elite Ironman^®^ 70.3 athletes, using machine learning (ML) algorithms. We collected the data of all professional triathletes competing between 2004 and 2020 in Ironman 70.3 races held worldwide. A sample of 16,611 professional athletes originating from 97 different countries and competing in 163 different races was thus obtained. Four different ML regression models were built, with gender, country of origin, and event location considered as independent variables to predict the final race time. For all the models, gender was the most important variable in predicting finish times. Attending to the single decision tree model, the fastest race times in the Ironman^®^ 70.3 World Championship of around ~4 h 03 min would be achieved by men from Austria, Australia, Belgium, Brazil, Switzerland, Germany, France, the United Kingdom, South Africa, Canada, and New Zealand. Considering the World Championship is the target event for most professional athletes, it is expected that training is planned so that they attain their best performance in this event.

## 1. Introduction

Triathlon is a multi-sports discipline. Long-distance triathlons, such as the ‘Ironman^®^ Hawaii’ (3.8 km swimming, 180 km cycling, and 42.195 km running), are known to many people [1] and have increased in popularity in terms of participants [2]. In a half-Ironman (‘Ironman^®^ 70.3’), the swimming, cycling, and running segments are half the distance of the corresponding segments in a full Ironman^®^ distance triathlon. Ironman^®^ ‘70.3’ refers to the total distance in miles (113.0 km) of the race, consisting of a 1.2 mile swim, a 56 mile bike ride, and a 13.1 mile run. Among different research topics involving endurance athletes, the interest in understanding the environmental characteristics related to the place of competition and country of origin of the athletes has increased over the last few years [3,4,5,6].

Regarding the country of origin of long-distance triathletes, it has been reported that most finishers, as well as the fastest athletes competing in ‘Ironman^®^ Hawaii’, originated from the United States of America [7]. Although the previous literature has showed the dominance of certain nations in specific sports disciplines [4,6,8,9], little evidence is available regarding the place of competition or if these characteristics are performance-related. As the most popular long-distance triathlon in the world, athletes travel hundreds of miles to compete in Ironman^®^ 70.3 events, with many travelling thousands of miles to compete at remote locations and achieve personal goals or to earn a spot in the World Championship.

As previously showed, travelling may have a negative effect on athletes’ performance [10], due to sleep disorders, nutritional and appetite changes, travel fatigue (i.e., hours spent seated and disruption in daily rhythm and routines). Considering these negative effects on athletes’ cognitive and physical levels, specific treatments to surpass these effects are necessary [10,11]. Thus, looking past the athletes’ country of origin, the characteristics and place of competitions can provide important information for athletes and coaches to better prepare and select the most appropriate race course in the world. Since no information is available regarding the role of the course characteristics in runners’ performance, this study intended to find the fastest race courses for elite Ironman^®^ 70.3 athletes, using machine learning (ML) algorithms for predictive statistics. Based on previous findings, we hypothesized that the USA would be the fastest country [7].

## 2. Materials and Methods

### 2.1. Ethical Approval

This study was approved by the Institutional Review Board of Kanton St. Gallen, Switzerland, with a waiver of the requirement for informed consent of the participants as the study involved the analysis of publicly available data (EKSG 01-06-2020).

### 2.2. Study Design, Sample, and Data Preparation

This is a cross-sectional study, using secondary data available and downloaded from the official Ironman^®^ website (www.ironman.com, accessed on 22 June 2022) using a Python script. We analyzed elite (PRO) finishers race data of all Ironman^®^ 70.3 races recorded in the Ironman website between 2004 and 2020. To register as an elite triathlete, an athlete must apply in the Ironman^®^ 70.3 community to become a “PRO” member with a copy of the athlete’s license status as a professional athlete from the respective national triathlon federation, and must also be affiliated with the International Triathlon Union (www.triathlon.org, accessed on 22 June 2022). Data obtained include the athletes’ sex and country of origin, the event location and year, and the race finish time. Exclusion criteria were as follows: (i) athletes who did not start or finish, (ii) disqualified athletes, (iii) records with missing split time, and (iv) inconsistent times (i.e., impossible split times or final times smaller than split times, etc.).

### 2.3. Statistical Analysis

Descriptive statistics are presented using mean and standard deviation, frequencies, and percentages. Four ML regression models were built and compared, using four different algorithms: random forest regressor, XG boost regressor, cat boot regressor, and decision tree regressor. All four algorithms are decision-tree-based. The simplest of all of them is the single decision tree algorithm, whilst the other three are tree-ensembles that use between a few hundred and a few thousand trees. For each algorithm, the mean absolute error (MAE) and R-squared (R^2^) metrics were calculated. In order to try and understand the algorithms’ logic, several model-agnostic explainability tools were used, including the algorithms’ features of relative importance, partial dependence plots (PDP), and the aggregated SHAP values. In addition, a graphical illustration of the single decision tree model is also provided. During the training, the model was fitted with 75% of the available data, and the remaining 25% was used to test and compare predictions vs. real values, calculating MAE and R^2^ as the accuracy metrics of the algorithm. The MAE presents the mean of the absolute values of the individual prediction errors over all instances in the test set, in which higher values mean higher prediction errors. R^2^ is a value between 0 and 1 and refers to the “goodness of fit”, that is, the model’s predictive power. The features’ relative importance represents a score for each feature in a specific model. Higher values correspond to higher importance in predicting the target variable. PDP and SHAP values provide information on how each feature contributes to the model output. Since the outcome (or predicted variable) was the finish time in seconds, the fastest courses were defined as the race locations where the average finish time was the lowest. The triathletes’ gender and country of origin, as well as the event location (race course) were used as predictors. As three of them were categorical variables, they were encoded into numerical values. Gender was encoded as 0 = female, 1 = male, whilst for the country and event location, a rank-encoding scheme based on the frequency was used. Data processing and analysis were performed using Python and associated ML libraries in a Jupyter notebook (Google Colab).

## 3. Results

A total of 16,611 athletes of both sexes from 97 different countries in 163 different race locations were sampled. Figure 1 presents the histograms of the finish times for both sexes. Based on the histograms, it is possible to see a higher frequency for men than women, and that men present a higher frequency of lower finish times.

Table 1 presents a comparative table with the performance scores (MAE and R^2^) and the features’ relative importance for all four predictive models considering the different algorithms and calculated using the 25% test sample. The XG boost and cat boost models were the models achieving the best predictive performance (R^2^ of 0.55), while the simplest of the models (model 4, single decision tree) obtained the lowest score (R^2^ of 0.43). In regards to the features’ importance, gender was the most important variable to predict the finish time for all five models. However, event location and country of origin presented different importance for different algorithms. For example, event location presented a higher importance compared to country when the random forest regressor was used, whereas for the decision tree, the triathlete’s country of origin was more important than the event location.

Figure 2 represents the aggregated SHAP values of each of the four models. The SHAP values show how each of the predictors contribute to the model output in respect to a reference value. For instance, the gender variable adds about 1000 s on average to the predicted finish time for females (blue points) and subtracts about 600 for males (red points). The relative low importance of the event location variable for the decision tree model can be appreciated in the respective chart, given the small size of its contributions to the model output. However, a few blue points can be seen at the far-left of the chart (between 0 and −500 s), indicating that locations with low values for *EventLocation_ID*, which have the highest number of participants, would be among the fastest race locations. Similarly, a few red dots at the far-right end of the XG boost and cat boost charts indicate that some countries with a low number of participants would be the worst-performing ones, adding 3000 s or more to the predicted race finish time.

Figure 3 and Figure 4 show the partial dependence plots (PDP) of each predictive model. The PDP plots provide additional details on the specific locations and/or countries that more contribute to the models’ predictions, as well as the amount of their contribution. Each of the charts show the contribution to the predictive model output for each value of each feature, with respect to a reference value of zero. For the gender predictor, 0 means females, whilst for country, a value of 0 corresponds to the USA, and for event location a value of 0 corresponds to the Ironman^®^ 70.3 World Championship.

All four models identify the Philippines as the worst-performing country (Country_ID = 13) adding around 2000 s on average (3000 in the case of the decision tree) to the finish time predictions with respect to the USA. Furthermore, the three ensemble algorithms identify several countries (IDs 2, 5, 9) as making negative contributions to the finish time predictions; however these contributions are relatively small (less than 1000 s) in all cases. The fourth model (single decision tree) shows a flatter country PDP chart which reflects the less complex nature of the algorithm.

In respect of the event location variable, the first three algorithms (the tree ensembles) show very similar charts with two distinctive lows at positions 35 and 37 corresponding to Ironman^®^ 70.3 Monterrey and Dubai, subtracting around 600–800 s from the output in respect to the reference (the World Championship). The last algorithm (single decision tree) shows a simpler chart, corresponding to its simpler logic, in which all and every single race location will add either 160, 200, or 220 s (approximately) to the reference value. According to the decision tree then, the Ironman^®^ 70.3 World Championship is the fastest race course.

Figure 4 represents the decision tree algorithm implemented as model 4. As already explained, all four predictive models are ensemble variations in the basic decision tree; hence, interpreting the latter is relevant to understanding all of them. Starting at node 0 (with gender encoded as 0 for women and 1 for men), the decision node *Gender_ID <= 0.5* leads to the women’s results on the top branch (node #1) and men´s at the bottom (node #16). Note the color coding in which the darker colors represent longer finish times (represented by the “value” in seconds). As we are searching for the best times, we continue to the men’s branch, in which decision node #16 *Country <= 11.5* splits the dataset into the first athletes’ countries by the number of records (i.e., Austria, Australia, Belgium, Brazil, Switzerland, Germany, France, United Kingdom, United States, South Africa, Canada, and New Zealand in node #17) and the rest (node #24). The algorithm tells us at this point that 47.8% of the records in the PRO sample were men from one of the first twelve countries by the number of participants, and that they achieved an average race time of 4 h 10 min (14,992.2 s). If we continue through node #18 (from node #17 when *EventLocation_ID < = 0.5*, that is, the Ironman^®^ World Championship) and then to node #20 (when *Country_ID > 0.5*, and is less or equal than 11 as determined earlier) we reach the fastest (whitest) node in this decision tree which corresponds to male triathletes from countries with IDs from 1 to 11 (Austria, Australia, Belgium, Brazil, Switzerland, Germany, France, United Kingdom, South Africa, Canada, and New Zealand) competing in the Ironman^®^ World Championship location and achieving an average time of 14,585.5 s (4 h 3 min) (Figure 5 and Figure 6).

## 4. Discussion

This study is the first to investigate the location of the fastest race course for Ironman^®^ 70.3 using ML algorithms. The first important finding was that the best race times in professional athletes in Ironman^®^ 70.3 competitions were achieved by men competing in the Ironman^®^ 70.3 World Championship. The World Championship is the target event for most professional athletes, so it is expected that athletes attain their best performance in this event. In addition, only the best-placed athletes in the Ironman^®^ 70.3 events can participate in the World Championship, which greatly raises the level of competition and competitiveness among the athletes. Since few studies have used a ML approach to predict performance or different outcomes in triathletes [12,13], comparisons are limited. Nevertheless, we expect that our data’s accuracy in the prediction of results could also guide future studies. Moreover, it is important to mention that the actual race course of the Ironman^®^ 70.3 World Championship changed between 2006 and 2020 (see Table 2). Regarding the average temperature (see Table 3), no fundamental differences exist between the various race courses; however, some differences according the month of the event as well as differences for the swimming, cycling, and running segments could be seen. These characteristics present some insights regarding the interplay between the intrapersonal (i.e., motivation), interpersonal (i.e., competitive environment), and environmental features (i.e., different race courses) for competitive athletes. Although these factors were beyond our scope in this study, future studies should investigate how these factors interact, and how performance can be conditioned by this interaction.

Another interesting finding was that the locations with the highest number of participants would be among the fastest race locations. Factors that explain these findings may be related to the race course, the difficulty in accessing the race event, weather characteristics, and personal motivation, since a larger number of athletes compete in these events. Since these are speculative, future studies should better explore these findings in order to achieve a deeper understanding of the influence of each of these effects.

Our study presents some advantages for performance prediction compared to other studies which have largely focused on the role of training [14], pacing [15,16], technical adjustments [16], and nutritional and morphological characteristics [17] in endurance athletes. The importance of the event location for athletes’ performance has not been sufficiently explored in the scientific literature [18,19]. Besides the influence of athletes and equipment characteristics on Ironman^®^ 70.3 triathlon performance [20,21], as an outdoor sport, the role of environmental characteristics needs to be considered, moving beyond the weather characteristics frequently studied (i.e., temperature, wind, humidity) [22]. In this way, our results indicate that the place of competition is an important variable to be considered for athletes to achieve their best performance. In addition, altitude, terrain characteristics, water temperature, sea current, and track characteristics influence performance, and are emergent topics that also need to be considered in the future.

Finally, all our predictive models showed that the best performance would be achieved by men. Despite the reduction in gender gaps in performance over the last few years, men still perform better than women [23]. Gender differences have been previously investigated [24,25] and performance differences range between 12% to 18%, depending on race characteristics [24]. In summary, morphological characteristics (e.g., lower limbs, body fat, body mass) and physiological determinants (e.g., maximal oxygen uptake, lactate threshold, movement economy) [23] were contextualized to explain the differences in the swimming, cycling, and running segments that reduce women’s performance [23]. Due to the large body of the literature available regarding sex performance differences, it is important to move beyond this topic. Future studies should investigate the role of the race course, in addition to the training background and intrapersonal motivation, in each group, as well as different performance levels and age categories. As mass sporting events have increased over the last few years and participants often compete in their leisure time, it would be desirable to obtain a good understanding of the impact of these race characteristics on their well-being.

It is acknowledged that this study is not free from limitations. The first important limitation was the lack of information regarding the race courses that could explain why the Ironman^®^ 70.3 World Championship presented the best characteristics for athletes’ performance, considering the different characteristics of the triathlon disciplines. The lack of information about the race course in each segment impaired our understanding of which discipline (swimming, cycling, running) would be the most positively affected by the race course characteristics and would have the strongest influence on the athletes’ finish times. Secondly, information regarding changes in race course characteristics over the years was not taken into consideration during the data analysis. The absence of additional information about the athletes’ personal characteristics, such as motivation, training background, and travel distance, also limits the generalization of these findings. However, our results bring valuable insights about the importance of the event location in performance and introduce some advances regarding the importance of statistical approaches in helping with athletes’ preparation.

## 5. Conclusions

Between 2004 and 2020, most professional Ironman^®^ 70.3 athletes achieved their fastest race times in the Ironman^®^ 70.3 World Championship with an average final time of around 4 h. Future studies should consider stratification based on the performance level, as well as understand the effects of racecourses in helping athletes achieve their best results.

## Figures and Tables

**Figure 1 ijerph-20-03619-f001:**
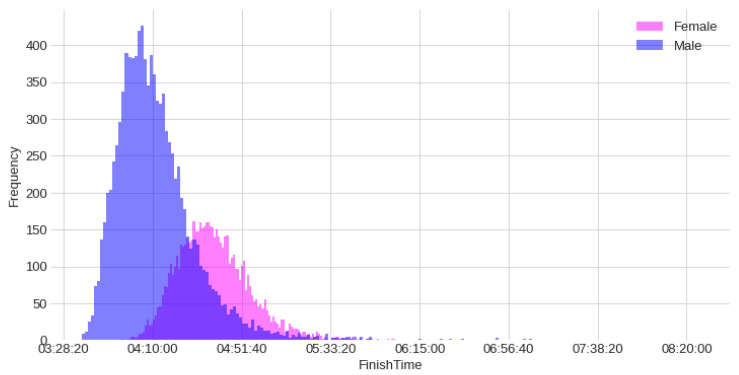
Histograms of the finish time of Ironman^®^ 70.3 PRO athletes competing between 2004 and 2020.

**Figure 2 ijerph-20-03619-f002:**
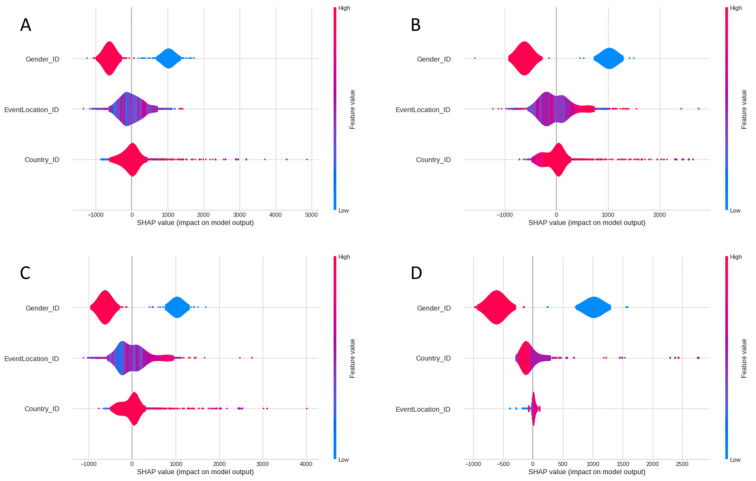
Aggregated SHAP values of the five predictive models: (**A**) Algorithm Random Forest Regressor; (**B**) Algorithm XG Boost Regressor; (**C**) Algorithm Cat Boot Regressor, (**D**) Algorithm Decision Tree Regressor).

**Figure 3 ijerph-20-03619-f003:**
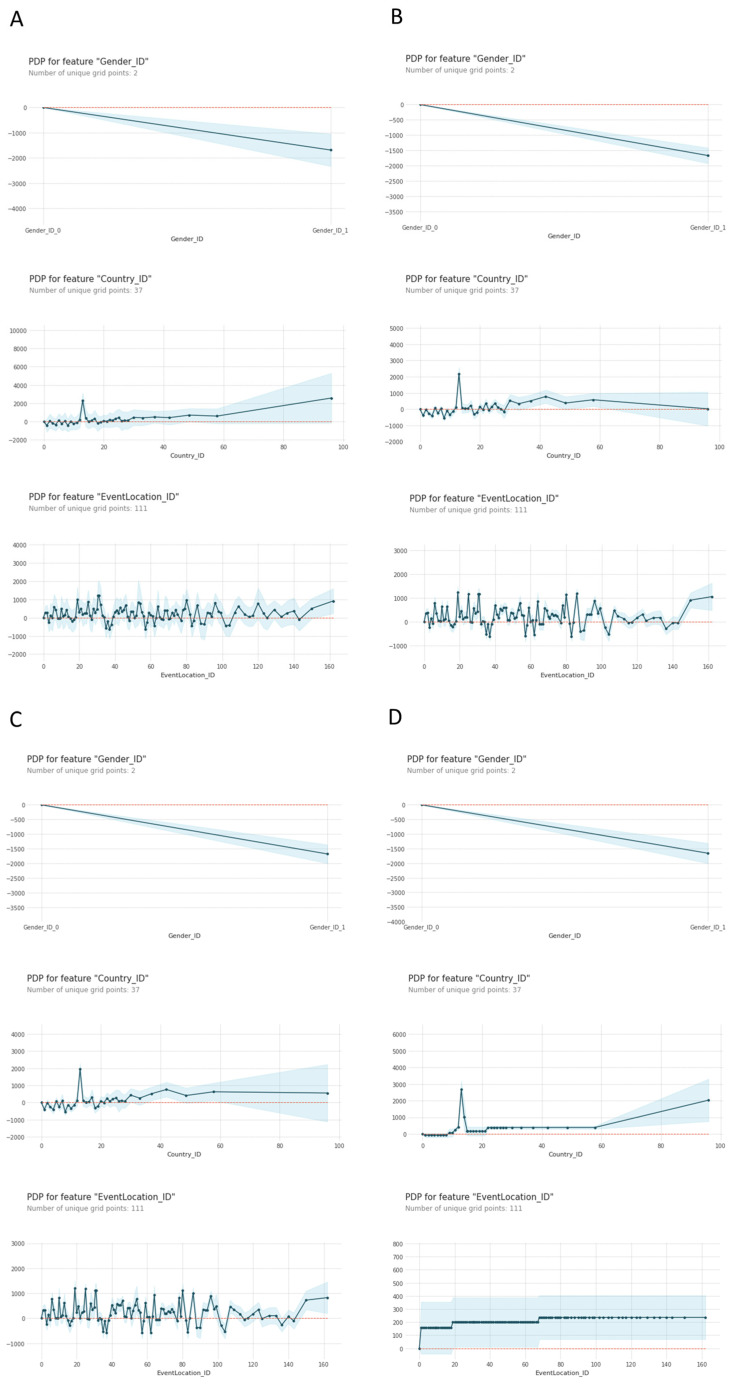
PDP plots of PDP plots of (**A**) Algorithm Random Forest Regressor, (**B**) Algorithm XG Boost Regressor; (**C**) Algorithm Cat Boost Regressor, and (**D**) Algorithm Decision Tree.

**Figure 4 ijerph-20-03619-f004:**
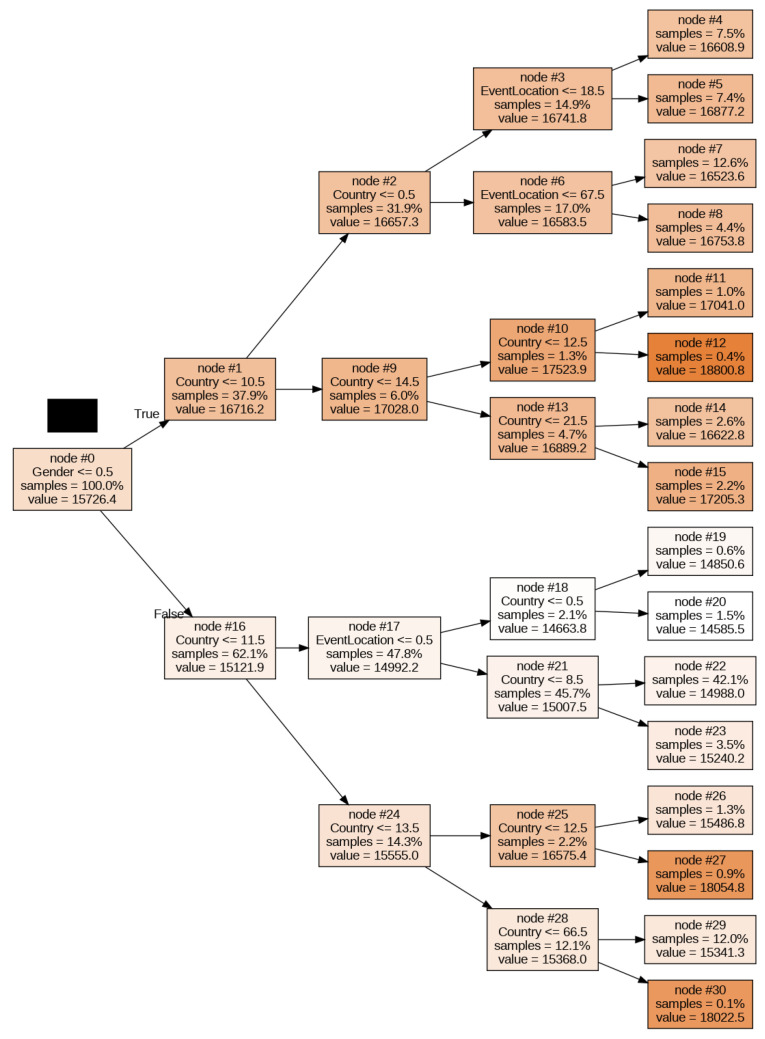
Decision tree (model 4) graphical representation.

**Figure 5 ijerph-20-03619-f005:**
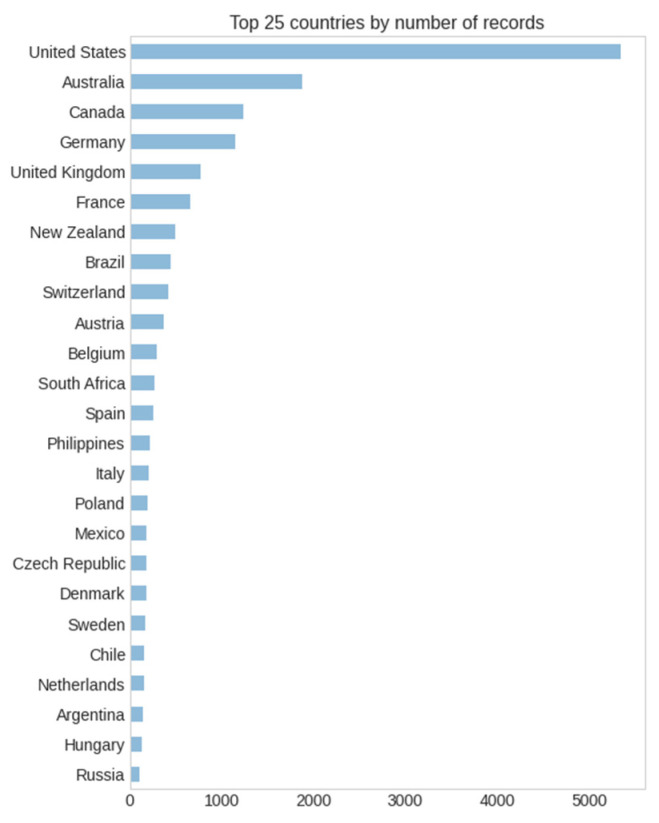
Top 12 countries by number of records.

**Figure 6 ijerph-20-03619-f006:**
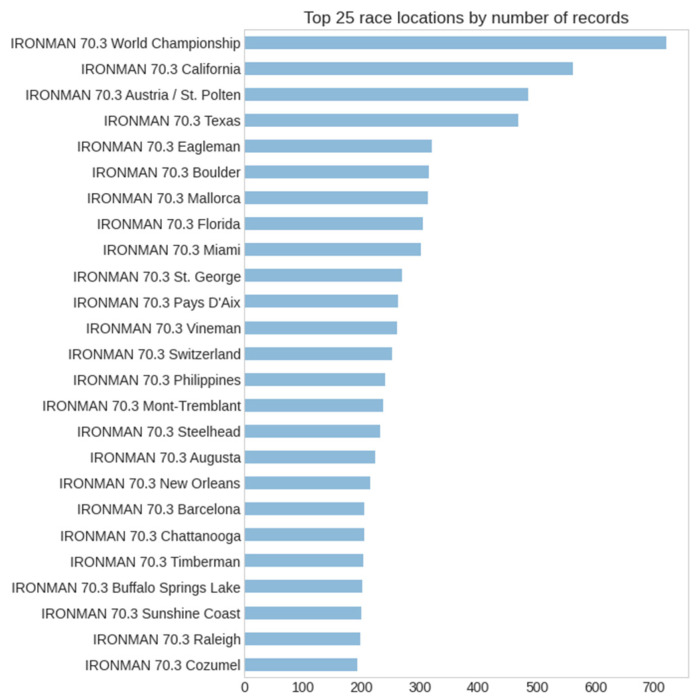
Top 10 locations by number of records.

**Table 1 ijerph-20-03619-t001:** Regression models results using the four different algorithms.

	Algorithm			
	Random Forest	XG Boost	Cat Boost	Decision Tree
MAE (seconds)	720.8	688.4	688.8	796.5
R-squared	0.51	0.55	0.55	0.43
**Feature (variable)**	Importance			
Gender	0.45	0.93	0.45	0.81
Event Location	0.32	0.04	0.30	0.01
Country	0.23	0.03	0.25	0.18

Legend: MAE (mean absolute error)

**Table 2 ijerph-20-03619-t002:** Locations of Ironman^®^ 70.3 World Championships.

Year	City, Country
2006–2010	Clearwater, FL, USA
2011–2013	Henderson, NV, USA
2014	Mont-Tremblant, QC, Canada
2015	Zell am See-Kaprun, Austria
2016	Mooloolaba, QLD, Australia
2017	Chattanooga, TN, USA
2018	Nelson Mandela Bay, South Africa
2019	Nice, Alpes-Maritimes, France
2020	Taupo, New Zealand

**Table 3 ijerph-20-03619-t003:** Specific descriptions of Ironman^®^ 70.3 race courses.

Race	Month	Average Temperature	Swimming Course Characteristics	Cycling Course Characteristics	Running Course Characteristics
IRONMAN^®^ 70.3 Zell am See	August	19 °C	1 lap of 1.9 km in a lake	1 lap of 90 km, altitude difference of 1270 m per lap, overall altitude difference of 1270 m	2 laps of 21.1 km, altitude difference of 26 m per lap, at an overall altitude difference of 52 m
IRONMAN^®^ 70.3 European Championship Elsinore	June	21 °C	1 lap of 1.9 km in a bay	1 lap of 90 km, altitude difference of 600 m per lap, overall altitude difference of 600 m	3.5 laps of 21.1 km, altitude difference of 28.6 m per lap, at an overall altitude difference 100 m
IRONMAN^®^ 70.3 Greece Costa Navarino	October	23 °C	1 lap of 1.9 km in the ocean	2 laps of 90 km, altitude difference of 540 m per lap, overall altitude difference of 1080 m	3 laps of 21.1 km, altitude difference of 66.7 m per lap, at an overall altitude difference 200 m
IRONMAN^®^ 70.3 Indian Wells La Quinta	December	18 °C	1 lap of 1.9 km in a lake	1 lap of 90 km, altitude difference of 161 m per lap, overall altitude difference of 161 m	2 laps of 21.1 km, altitude difference of 57.7 m per lap, at an overall altitude difference of 115 m
IRONMAN^®^ 70.3 Middle East Championship Bahrain	December	22 °C	1 lap of 1.9 km in a bay	1 lap of 90 km, altitude difference of 300 m per lap, overall altitude difference of 300 m	3 laps of 21.1 km, altitude difference of 24.3 m per lap, at an overall altitude difference 73 m
IRONMAN^®^ 70.3 Texas	April	21 °C	1 lap of 1.9 km in a bay	1 lap of 90 km, altitude difference of 45 m per lap, overall altitude difference of 45 m	3 laps of 21.1 km, altitude difference of 10.7 m per lap, at an overall altitude difference 32 m
IRONMAN^®^ 70.3 Tallinn	August	17 °C	1 lap of 1.9 km in a lake	1 lap of 90 km, altitude difference of 300 m per lap, overall altitude difference of 300 m	2 laps of 21.1 km, altitude difference of 57.7 m per lap, at an overall altitude difference of 115 m
IRONMAN^®^ 70.3 Western Sydney	September	23 °C	1 lap of 1.9 km in a lake	2 laps of 90 km, altitude difference of 165 m per lap, overall altitude difference of 330 m	1 lap of 21.1 km, altitude difference of 25 m per lap, at an overall altitude difference of 25 m
IRONMAN^®^ 70.3 Maceio	August	28 °C	1 lap of 1.9 km in the ocean	1 lap of 90 km, altitude difference of 200 m per lap, overall altitude difference of 200 m	3 laps of 21.1 km, altitude difference of 23.3 m per lap, at an overall altitude difference 70 m
IIRONMAN^®^ 70.3 Liuzhou	September	28 °C	1 lap of 1.9 km in a river	2 laps of 90 km, altitude difference of 168.5 m per lap, overall altitude difference of 337 m	2 laps of 21.1 km, altitude difference of 171 m per lap, at an overall altitude difference of 342 m
IRONMAN^®^ 70.3 Luxembourg	June	18 °C	1 lap of 1.9 km in a river	1 lap of 90 km, altitude difference of 580 m per lap, overall altitude difference of 580 m	2,5 laps of 21.1 km, altitude difference of 40 m per lap, at an overall altitude difference of 100 m
IRONMAN^®^ 70.3 Marbella	May	21 °C	1 lap of 1.9 km in the ocean	1 lap of 90 km, altitude difference of 1400 m per lap, overall altitude difference of 1400 m	2 laps of 21.1 km, altitude difference of 25 m per lap, at an overall altitude difference of 50 m
IRONMAN^®^ 70.3 Vichy	August	28 °C	1 lap of 1.9 km in a lake	1 lap of 90 km, altitude difference of 1000 m per lap, overall altitude difference of 1000 m	2 laps of 21.1 km, altitude difference of 50 m per lap, at an overall altitude difference of 100 m
IRONMAN^®^ 70.3 Gdynia	August	25 °C	1 lap of 1.9 km in a bay	1 lap of 90 km, altitude difference of 1860 m per lap, overall altitude difference of 1860 m	2 laps of 21.1 km, altitude difference of 70 m per lap, at an overall altitude difference of 340 m
IRONMAN^®^ 70.3 Mallorca	May	25 °C	1 lap of 1.9 km in the ocean	1 lap of 90 km, altitude difference of 850 m per lap, overall altitude difference of 850 m	3 laps of 21.1 km, altitude difference of 20 m per lap, at an overall altitude difference 60 m
IRONMAN 70.3 Panama	March	30 °C	1 lap of 1.9 km in the ocean	3 laps of 90 km	3 laps of 21.1 km
IRONMAN 70.3 Augusta	September	27 °C	1 lap of 1.9 km in the river	1 lap of 90 km	2 laps of 21.1 km
IRONMAN 70.3 North Carolina	October	18 °C	1 lap of 1.9 km in the ocean	1 lap of 90 km	1 lap of 21.1 km
IRONMAN 70.3 Dubai	March	24 °C	1 lap of 1.9 km in the ocean	1 lap of 90 km, altitude difference of 87 m per lap, overall altitude difference of 87 m	1,5 laps of 21.1 km, altitude difference of 5 m per lap, at an overall altitude difference 7.5 m
IRONMAN 70.3 Sao Paolo	September	25 °C	1 lap of 1.9 km in a reservoir	2 laps of 90 km, altitude difference of 20 m per lap, overall altitude difference of 40 m	3 laps of 21.1 km, altitude difference of 15 m per lap, at an overall altitude difference 45 m
IRONMAN 70.3 Alagolas	August	28 °C	1 lap of 1.9 km in the ocean	1 lap of 90 km, altitude difference of 100 m per lap, overall altitude difference of 100 m	3 laps of 21.1 km, altitude difference of 20 m per lap, at an overall altitude difference 60 m
IRONMAN 70.3 Emilia Romagna	September	25 °C	1 lap of 1.9 km in the ocean	1 lap of 90 km, altitude difference of 185 m per lap, overall altitude difference of 185 m	3 laps of 21.1 km, altitude difference of 5 m per lap, at an overall altitude difference 15 m
IRONMAN 70.3 Pays D’Aix	May	20 °C	1 lap of 1.9 km in a lake	1 lap of 90 km, altitude difference of 390 m per lap, overall altitude difference of 390 m	3 laps of 21.1 km, altitude difference of 15 m per lap, at an overall altitude difference 45 m
IRONMAN 70.3 Sunshine Coast	September	26 °C	1 lap of 1.9 km in the ocean	2 laps of 90 km, altitude difference of 35 m per lap, overall altitude difference of 70 m	2 laps of 21.1 km, altitude difference of 20 m per lap, at an overall altitude difference 40 m
IRONMAN 70.3 Geelong	March	19 °C	1 lap of 1.9 km in a bay	2 laps of 90 km, altitude difference of 75 m per lap, overall altitude difference of 150 m	2,5 laps of 21.1 km, altitude difference of 25 m per lap, at an overall altitude difference 67 m
IRONMAN 70.3 Steelhead	Juna	20 °C	1 lap of 1.9 km in a lake	1 lap of 90 km	2 laps of 21.1 km
IRONMAN 70.3 Turkey	November	24 °C	2 laps of 1.9 km in the ocean	2 laps of 90 km, altitude difference of 20 m per lap, overall altitude difference of 40 m	3 laps of 21.1 km, altitude difference of 5 m per lap, at an overall altitude difference 15 m
IRONMAN 70.3 California	April	17 °C	1 lap of 1.9 km in the ocean	1 lap of 90 km, altitude difference of 220 m per lap, overall altitude difference of 220 m	2 laps of 21.1 km, altitude difference of 10 m per lap, at an overall altitude difference 20 m
IRONMAN 70.3 Astana	June	25 °C	1 lap of 1.9 km in a river	1 lap of 90 km, altitude difference of 20 m per lap, overall altitude difference of 20 m	2 laps of 21.1 km, altitude difference of 5 m per lap, at an overall altitude difference 10 m
IRONMAN 70.3 Florianapolis	April	21 °C	1 lap of 1.9 km in the ocean	1 lap of 90 km	3 laps of 21.1 km

## Data Availability

The data were downloaded from the official Ironman website (www.ironman.com, accessed on 22 June 2022).

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
