# Peer review of "A Machine Learning Approach to Finding the Fastest Race Course for Professional Athletes Competing in Ironman® 70.3 Races between 2004 and 2020"

_ijerph, 2023, doi:10.3390/ijerph20043619_

Round 1
Reviewer 1 Report
Dear authors,
Overall, this is an important article related to endurance performance using machine learning as a tool to predict performance. In my point of view, this is a trend that may help sports scientists and professionals worldwide.
Congrats on the idea of the article. I have only some questions, suggestions, and comments that could improve the quality of the article. I hope that they do sound not bad to you because my intention was only to help the authors to think better about some important topics for the study.
As a major review, I think it would be important for the authors to reply to the following questions:
What about genders? It was one of the most important variables in predicting the finish time in the study.
And the age? Is it important for 70.3 performance? Why you did not consider age in the ML model?
Were the split times assessed? Why they weren´t included in the ML model?
What is the author´s hypothesis?
Concerning the minor review, the specific comments have been described below:
Abstract:
Lines 21-22: In my point of view, you should insert machine learning into the aim of the study.
Line 24: Add “machine learning (ML)” abbreviation, please.
Line 33: I suggest changing the term “their peak performance” to “their best performance”.
Introduction:
Lines 42 and 43: This part is confusing and could be removed. I didn´t understand why the authors described the distance in both miles and kilometers. I suggest rewriting this part and removing the distance in miles.
Lines 45-59: In my opinion, this paragraph should be reformulated. The data does not allow the authors to assess the effect of the travel and this point is not related to the aim of the study. This part requires an adjustment or should be removed. I suggest the authors develop a rationale based on training preparation, the expensive price to travel and participate in all races, and especially that the athletes could know their higher probability to reach success.
Materials and Methods: What about the study type or study design? This information could be added at the start of the section.
Line 106: A better explanation about country and event location is required because they can sound the same thing. For instance, the country where the race occurred or the athletes´ nationality? It is not clear in the text.
Results:
In the results, I understood that “country” was the place where the race occurred, but it must be clarified and explained better in the methods section.
Discussion:
Why tables 2 and 3 appeared only in the discussion section? In my point of view, they aren´t the core of the study. If necessary, the authors would add them as a supply material. Sincerely, I didn´t understand how they can contribute to the study´s comprehension.
In the opinion of this reviewer, the split times may have a deep discussion.
Author Response
#REVIEWER 1
Dear authors,
Overall, this is an important article related to endurance performance using machine learning as a tool to predict performance. In my point of view, this is a trend that may help sports scientists and professionals worldwide.
Congrats on the idea of the article. I have only some questions, suggestions, and comments that could improve the quality of the article. I hope that they do sound not bad to you because my intention was only to help the authors to think better about some important topics for the study.
As a major review, I think it would be important for the authors to reply to the following questions:
What about genders? It was one of the most important variables in predicting the finish time in the study.
Authors answer: We intended to develop several ML predictive models that would use known information such as the tri-athlete´s gender and country of origin, and the race location, to predict the tri-athlete’s performance. Whilst the gender and the country of origin are not variables that a tri-athlete could influence, we desired to know if the choice of race location would be relevant. But naturally, and expected to a large extent, when we investigated the models feature importance, gender came up as the most relevant feature among the three predictors.
And the age? Is it important for 70.3 performance? Why you did not consider age in the ML model?
Authors answer: This study considered only PRO tri-athletes and unfortunately age information was not registered for them in the database. We are in the process of conducting additional studies for amateur triathletes where the age information is available, so age will also be used as a predicting variable.
Were the split times assessed? Why they weren´t included in the ML model?
Authors answer: Including the split times in a predictive model (even a simple multiple linear regressor) of the race time would achieve a perfect fit (R2=1) and this would completely mask the much weaker influence of other factors such as the country of origin or the race location. So, it was not right for this study.
What is the author´s hypothesis?
Authors answer: Included
Concerning the minor review, the specific comments have been described below:
Lines 21-22: In my point of view, you should insert machine learning into the aim of the study.
Authors answer: Adjusted
Line 24: Add “machine learning (ML)” abbreviation, please.
Authors answer: Adjusted
Line 33: I suggest changing the term “their peak performance” to “their best performance”.
Authors answer: Adjusted
Introduction:
Lines 42 and 43: This part is confusing and could be removed. I didn´t understand why the authors described the distance in both miles and kilometers. I suggest rewriting this part and removing the distance in miles.
Authors answer: Adjusted. Since that 70.3 refer to the total distance in miles, we decide to maintain in mile.
Lines 45-59: In my opinion, this paragraph should be reformulated. The data does not allow the authors to assess the effect of the travel and this point is not related to the aim of the study. This part requires an adjustment or should be removed. I suggest the authors develop a rationale based on training preparation, the expensive price to travel and participate in all races, and especially that the athletes could know their higher probability to reach success.
Authors answer: We appreciate your comment. We adjusted the paragraph.
Materials and Methods: What about the study type or study design? This information could be added at the start of the section.
Authors answer: Adjusted
Line 106: A better explanation about country and event location is required because they can sound the same thing. For instance, the country where the race occurred or the athletes´ nationality? It is not clear in the text.
Authors answer: We appreciate the suggestion. We tried to clarify this information in methods section, and also during the description of the results.
Results:
In the results, I understood that “country” was the place where the race occurred, but it must be clarified and explained better in the methods section.
Authors answer: We have specified in the methods section, and wherever else it could be ambiguous, that we are referring to the tri-athlete´s country of origin.
Discussion:
Why tables 2 and 3 appeared only in the discussion section? In my point of view, they aren´t the core of the study. If necessary, the authors would add them as a supply material. Sincerely, I didn´t understand how they can contribute to the study´s comprehension.
Authors answer: We agree with the expert reviewer. However, to show that differences exist between the editions we added in the limitations ‘As mentioned, the race course of the World Championship changed annually after Clearwater (2006-2010) and Henderson (2011-2013) (Table 2) and the course profiles are different for each race (Table 3).
In the opinion of this reviewer, the split times may have a deep discussion.
Authors answer: In a previous comment we present an answer about the exclusion of the split time in our predictive models. To avoid noise, we excluded this information from the methods section.
Reviewer 2 Report
The authors used a decision tree and tree-based ensemble algorithms to predict the final race time of the Ironman Championship. It is a well-written and easy-to-follow paper.
However, the paper includes lots of missing information and requires revisions. Here are my concerns:
- The abstract does not present adequate information. It would be better if the authors revised it to explain the work and conclude the remarks.
- Machine learning studies generally require more than one evaluation metric. The authors considered the MAE to evaluate the results of the non-normalized data. The results clearly indicate the superior model; however, the MAE minimizes the higher prediction errors. Considering the MSE and coefficient of determination (R2 score) would clarify which model is superior and more effective. In addition, Table 1 presents that Cat Boost obtains the minimum MAE, but the authors focused on the decision tree while presenting results.
- The paper includes no information about the parameters used for the algorithms. For example, which criterion (i.e., Gini Index) is used for the decision tree? How many trees were created for the random forest? etc.
- Figure 1 shows that the men complete the event in a short time. Therefore, it is easy for ML algorithms to determine gender as the most important factor. Unfortunately, as discussed by the authors, the dataset limits the study. However, Table 3 presents the average temperature for different championships, and investigating the effect of the average temperature on the athletes' performances from different countries might give a more significant finding.
- Detailing the experiments and presenting more concrete findings would make the study more useful.
Author Response
#REVIEWER 2
The authors used a decision tree and tree-based ensemble algorithms to predict the final race time of the Ironman Championship. It is a well-written and easy-to-follow paper.
However, the paper includes lots of missing information and requires revisions. Here are my concerns:
- The abstract does not present adequate information. It would be better if the authors revised it to explain the work and conclude the remarks.
Authors answer: We have updated some narrative to the ‘Abstract’ section highlighting the use of machine learning algorithmics and the reasoning why.
- Machine learning studies generally require more than one evaluation metric. The authors considered the MAE to evaluate the results of the non-normalized data. The results clearly indicate the superior model; however, the MAE minimizes the higher prediction errors. Considering the MSE and coefficient of determination (R2 score) would clarify which model is superior and more effective. In addition, Table 1 presents that Cat Boost obtains the minimum MAE, but the authors focused on the decision tree while presenting results.
Authors answer: We do recognize the reviewer´s advice and have re-run the code adding the R2 metric. Just for curiosity and completion. In this study, which is the first in a series, we had only aimed to get some degree of interpretability of the decision-tree based algorithms, and to draw some conclusions on the machine learning algorithms logic that could provide valuable insights. PS: after adding the R2 metric calculations I must admit a couple of the algorithms, in particular Cat Boost and XG Boost were the best performers in terms of accuracy with an R2 of 0.55. And this is not a bad R2 score for a model with 3 variables (although for a limited PRO population).
- The paper includes no information about the parameters used for the algorithms. For example, which criterion (i.e., Gini Index) is used for the decision tree? How many trees were created for the random forest? etc.
Authors answer: The tuning of the algorithms (things like number of estimators / trees for ensembles, or the max depth parameter of the decision tree) has been done manually, given the small sample size allowed for short computation times (typically a few seconds in most cases and never over 30 seconds for the PRO subset). The decision tree used 5 levels of depth, whilst among the ensemble algorithms the number of estimators varied from 100 to 2 or 3 thousand. But then, we also have the learning rate parameter and some more, so I accepted a few defaults to start with. Gini was default with decision tree regressors indeed.
See below from the sklearn libray documentation:
feature_importances_
The importance of a feature is computed as the (normalized) total reduction of the criterion brought by that feature. It is also known as the Gini importance.
Warning: impurity-based feature importances can be misleading for high cardinality features (many unique values). See sklearn.inspection.permutation_importance as an alternative.
Returns:
feature_importances_ndarray of shape (n_features,)
Normalized total reduction of criteria by feature (Gini importance).
But then we felt going too deep into all this was unnecessary in an article about Ironman 70.3
- Figure 1 shows that the men complete the event in a short time. Therefore, it is easy for ML algorithms to determine gender as the most important factor. Unfortunately, as discussed by the authors, the dataset limits the study. However, Table 3 presents the average temperature for different championships, and investigating the effect of the average temperature on the athletes' performances from different countries might give a more significant finding.
Authors answer: The temperature data was not available in the dataset, so it could not be used. The average temperature given in the table is just for reference.
- Detailing the experiments and presenting more concrete findings would make the study more useful.
Authors answer: We have improved the narrative throughout the article to better describe the objectives and methods, whilst at the same time acknowledge the data available was limited.

Round 2
Reviewer 2 Report
Thanks to the authors for considering my concerns and revising the paper. Almost all of my comments are addressed in the paper.
Still, I have only one concern:
Discussing and presenting the decision tree results with more focus is to defend the findings of a model of lower prediction ability than the results produced by the superior model. This shows that the results could be misinterpreted in real life. For example, it could not be suitable to consider the characteristics of the bronze medalist when you have a gold medalist in your team.
My recommendation is to analyze all results obtained by all models (the authors improved this in the revised version) and not make focus on the decision tree. This might be done by separating the results section into subsections for each model and discussing all the results in the discussion section.
Author Response
Reviewer 2
Thanks to the authors for considering my concerns and revising the paper. Almost all of my comments are addressed in the paper.
Still, I have only one concern:
Discussing and presenting the decision tree results with more focus is to defend the findings of a model of lower prediction ability than the results produced by the superior model. This shows that the results could be misinterpreted in real life. For example, it could not be suitable to consider the characteristics of the bronze medalist when you have a gold medalist in your team.
Answer: we changed as suggested
My recommendation is to analyze all results obtained by all models (the authors improved this in the revised version) and not make focus on the decision tree. This might be done by separating the results section into subsections for each model and discussing all the results in the discussion section.
Answer: we changed as suggested